# Specific S100 Proteins Bind Tumor Necrosis Factor and Inhibit Its Activity

**DOI:** 10.3390/ijms232415956

**Published:** 2022-12-15

**Authors:** Alexey S. Kazakov, Marina Y. Zemskova, Gleb K. Rystsov, Alisa A. Vologzhannikova, Evgenia I. Deryusheva, Victoria A. Rastrygina, Andrey S. Sokolov, Maria E. Permyakova, Ekaterina A. Litus, Vladimir N. Uversky, Eugene A. Permyakov, Sergei E. Permyakov

**Affiliations:** 1Institute for Biological Instrumentation, Pushchino Scientific Center for Biological Research of the Russian Academy of Sciences, 142290 Pushchino, Russia; 2G.K. Skryabin Institute of Biochemistry and Physiology of Microorganisms, Pushchino Scientific Center for Biological Research of the Russian Academy of Sciences, 142290 Pushchino, Russia; 3Department of Molecular Medicine and USF Health Byrd Alzheimer’s Research Institute, Morsani College of Medicine, University of South Florida, Tampa, FL 33612, USA

**Keywords:** cytokine, tumor necrosis factor, S100 protein, protein–protein interaction, inflammatory diseases

## Abstract

Tumor necrosis factor (TNF) inhibitors (anti-TNFs) represent a cornerstone of the treatment of various immune-mediated inflammatory diseases and are among the most commercially successful therapeutic agents. Knowledge of TNF binding partners is critical for identification of the factors able to affect clinical efficacy of the anti-TNFs. Here, we report that among eighteen representatives of the multifunctional S100 protein family, only S100A11, S100A12 and S100A13 interact with the soluble form of TNF (sTNF) in vitro. The lowest equilibrium dissociation constants (*K_d_*) for the complexes with monomeric sTNF determined using surface plasmon resonance spectroscopy range from 2 nM to 28 nM. The apparent *K_d_* values for the complexes of multimeric sTNF with S100A11/A12 estimated from fluorimetric titrations are 0.1–0.3 µM. S100A12/A13 suppress the cytotoxic activity of sTNF against Huh-7 cells, as evidenced by the MTT assay. Structural modeling indicates that the sTNF-S100 interactions may interfere with the sTNF recognition by the therapeutic anti-TNFs. Bioinformatics analysis reveals dysregulation of TNF and S100A11/A12/A13 in numerous disorders. Overall, we have shown a novel potential regulatory role of the extracellular forms of specific S100 proteins that may affect the efficacy of anti-TNF treatment in various diseases.

## 1. Introduction

Tumor necrosis factor (TNF), also known as TNF-α, TNF ligand superfamily member 2 or Cachectin, is a potent pleiotropic (pro/anti)-inflammatory cytokine, existing in a soluble form (sTNF) that acts as a ligand, or a membrane-bound form (mTNF, single pass type II membrane protein) that acts as either a ligand or a receptor (reviewed in refs. [1,2,3,4]). The gene of full-length human mTNF (233 amino acid residues) is rapidly transcribed in macrophages, B/T-cells, Langerhans cells, dendritic cells, etc. [5], upon exposure to pathogens or signals of inflammation/stress [6]. The proteolytic cleavage of mTNF by Disintegrin and metalloproteinase domain-containing protein 17 releases sTNF (157 residues) [7]. sTNF is an inactive monomer at the picomolar protein level, and rapidly forms an active non-covalent trimer at nanomolar concentrations [8]. The trimer is characterized by edge-to-face packing of the antiparallel β-sandwiches of the individual monomers with the ‘jelly roll’ structure characteristic of the viral capsid proteins (TNF-like fold, SCOP [9] ID: 2000041) [10]. Both sTNF and mTNF signal through type I transmembrane receptors TNF-R1 (CD120a, p55, p60) and TNF-R2 (CD120b, p75) [2], but TNF-R2 is more efficiently activated by mTNF [11]. TNF-R1 is ubiquitously expressed in the body, whereas TNF-R2 expression is mainly restricted to the Langerhans cells, dendritic cells, macrophages, Kupffer cells, monocytes, and T-cells [5]. While sTNF can act at the sites distant from the TNF-producing cells, mTNF acts locally as a ligand in a cell-to-cell contact manner [12]. Besides, as a receptor, mTNF transmits the reverse (‘outside-to-inside’) signals back to the cells upon binding to its native receptors [2]. Both TNF-R1 and TNF-R2 can also be converted by proteolytic cleavage to the soluble forms that operate as decoy receptors [13]. TNF-R1/R2 are structurally similar in their extracellular regions but differ in the intracellular regions, leading to the divergence of the adaptor proteins engaged in the downstream signaling [3]. While both TNF receptor-mediated signaling pathways may activate the prosurvival transcription factors NF-κB and AP1, TNF-R1 is also able to trigger a cell death response (apoptosis or necroptosis and inflammation) depending on the physiological conditions [3]. Several hundred genes are regulated by TNF in a cell type-specific manner with distinct patterns of induction kinetics, which ultimately leads to changes in the cellular responses to stimuli [14,15].

The complicated regulation of TNF signaling and the wide range of the cellular events evoked by TNF seem to explain its association with progression of numerous disorders, including inflammatory diseases (rheumatoid arthritis, inflammatory bowel disease, plaque psoriasis, psoriatic arthritis, ankylosing spondylitis, juvenile idiopathic arthritis, uveitis), pulmonary diseases (asthma, chronic bronchitis, chronic obstructive pulmonary disease, acute lung injury and acute respiratory distress syndrome), metabolic disorders (obesity, type 2 diabetes mellitus), cancer, cardiovascular disease, infectious and neurodegenerative diseases, fibrosis and Dupuytren’s disease, renal disease and hypernociception [14,16,17,18,19,20]. Etanercept, infliximab, adalimumab, certolizumab pegol and golimumab are approved monoclonal antibody-based anti-TNFs for treatment of rheumatoid arthritis, Crohn’s disease, juvenile idiopathic arthritis, psoriatic arthritis, axial spondyloarthritis, plaque psoriasis, ulcerative colitis, enthesitis-related arthritis, hidradenitis suppurativa, uveitis, ankylosing spondylitis [21,22]. There are also some off-label indications for them [21]. Anti-TNF therapy appears to be protective in severe COVID-19 [23]. Recombinant sTNF itself (tasonermin) is approved for irresectable soft tissue sarcoma of the limb [24].

The clinical success of TNF inhibitors (the global market exceeds USD 38 billion [25]) drives development of numerous next-generation TNF-modulating therapeutics based on more precise knowledge of the regulatory aspects of TNF signaling [1,17,22]. In this regard, identification of the extracellular TNF binding partners that can affect therapeutic efficacy of the TNF-modulating drugs becomes critical for their successful clinical application. To this end, here we explore the possibility of regulatory activity of extracellular forms of S100 proteins with respect to sTNF, based on the recent data on multiple interactions of specific S100 proteins with some cytokines.

The S100 protein family contains over 20 structurally homologous, but functionally diverse, Ca^2+^-binding proteins of the EF-hand superfamily, some of which also bind Zn^2+^/Cu^2+^/Fe^2+^ (reviewed in ref. [26,27,28,29]). S100 proteins exhibit cell/tissue-specific expression and may localize in the cytosol, nucleus and extracellular space. Their broad spectrum of functional activities is modulated by metal-binding, multimerization, posttranslational modifications and interaction with various targets (receptors, ion channels, cytoskeletal proteins, enzymes, transcription factors, cytokines, nucleic and fatty acids). S100 proteins regulate multiple vital cellular processes, including cell differentiation/proliferation/death, energy metabolism, metal homeostasis, inflammation, pathogen resistance, etc. They are associated with a wide range of oncological, cardiovascular, respiratory, neurological, inflammatory and autoimmune diseases [28,30,31,32,33,34,35] that overlap with the spectrum of TNF-related diseases (see above). When released into the extracellular space, some S100 proteins act in a cytokine-like manner, recognizing cell surface receptors: RAGE, TLR4, ErbB1, ErbB3, ErbB4, CD36, CD68, CD147, CD166, neuroplastin-β, 5-HT_1B_, IL-10R and SIRL-1 [29,36,37]. Furthermore, some of the extracellular S100 proteins may affect cytokine signaling via direct binding of various cytokines. The most promiscuous S100 protein, S100P, interacts with a large set of the four-helical cytokines, such as CLCF1, CNTF, CT-1, EPO, G-CSF, GH, GH-V, GM-CSG, IFN-β, IFN-ω1, IL-3/5/9/10/11/13/15/20/21/22/24/26/27/31/35, LEP, OSM, PRL and THPO [36,38,39,40,41,42]. S100A1/A6 interact with CLCF1, CNTF, CT-1, IFN-β and IL-11 [42,43]. S100A6 also binds to EPO along with S100A2 [41]. S100A4 interacts with ErbB1 ligands and IFN-β [43,44]. S100A11 was shown to bind MIF [45]. S100A13 interacts with C-C motif chemokine 5, FGF1 and IL-1α [46,47,48]. S100B protein binds FGF2, IFN-β and IL-11 [42,49,50,51].

The extensive interactions between several S100 proteins and specific cytokines, including those belonging to the all-beta class (the case of TNF), and a significant overlap in the diseases associated with disturbance of S100 proteins or TNF, indicate the possibility of direct or indirect relationships between them. In the present work, we show that Ca^2+^-bound forms of specific S100 proteins are highly specific for sTNF and inhibit its activity against human hepatoma Huh-7 cells.

## 2. Results and Discussion

### 2.1. Surface Plasmon Resonance Study of sTNF Recognition by S100 Proteins

To probe the affinity of the sTNF monomer to the S100 protein family representatives at 25 °C, recombinant human sTNF (3 μM) was immobilized on the surface of the SPR sensor chip by amine coupling, flushed with the running buffer for two days and several times with 10 mM glycine pH 3.3 for 150 s to ensure dissociation of sTNF multimer(s) [52,53], and solutions of Ca^2+^-loaded forms of the bacterially expressed human S100A1/A2/A3/A4/A5/A6/A7/A8/A9/A10/A11/A12/A13/A14/A15/A16/B/P proteins (up to 2 µM) were passed over the chip. Most of the studied S100 proteins showed no signs of specificity for sTNF, which suggests that the corresponding dissociation constants (*K_d_*) exceed 10^−4^ M. Meanwhile, the SPR sensograms for S100A11/A12/A13 proteins reveal the analyte concentration-dependent effects (Figure 1). The SPR data are adequately described using the heterogeneous ligand model (see Section 3.2) (Figure 1), which was earlier shown to be suitable for the description of the interactions between several S100 proteins and cytokines [36,38,40,41,42,43,49,51]. The lowest *K_d_* values for the complexes with the monomeric sTNF range from 2 nM (S100A12/A13) to 28 nM (S100A11) (Table 1). Importantly, the *K_d_* values reach the levels of extracellular forms of the S100 proteins under physiological conditions, indicating that the S100–sTNF interactions may be physiologically significant. For example, the mean S100A11 concentration in the synovial fluid of patients with rheumatoid arthritis is 17 nM, and the maximal protein concentrations reach 84 nM [54]. The average S100A12 level in serum/plasma of 5–9 nM raises under the pathological conditions to 24–688 nM [55,56,57,58]. The mean S100A12 concentration in the synovial fluid in the rheumatoid arthritis is 179 nM [59]. Finally, the highest S100A13 level in the synovial fluid at a microfracture reaches 0.6 nM [60]. Furthermore, local concentrations of extracellular forms of the S100 proteins in the damaged S100-expressing tissues should be even higher, further promoting the S100–sTNF interactions. Meanwhile, the estimates of the mean basal blood and cerebrospinal fluid concentrations of sTNF typically do not exceed 0.3 pM [61,62] and 0.7 pM [63], respectively. Under pathological conditions, these values raise to 0.9 pM (breast cancer [61]) and 3/41 pM (subarachnoid hemorrhage [63]/bacterial meningitis [64]), respectively. The mean sTNF concentration in the synovial fluid of the patients with rheumatoid arthritis is 19 pM [65]. Thus, the sTNF levels are generally orders of magnitude lower than those for the extracellular S100 proteins, indicating that sTNF binding to the S100 proteins probably cannot affect signaling of the latter. On the contrary, association of the S100 proteins with sTNF could alter signaling of this cytokine, as exemplified by the IFN-β interaction with S100A1/A4/B/P proteins [36,43,49].

The only reported equilibrium dimer dissociation constant for Ca^2+^-bound forms of S100A11/A12/A13 proteins of 160 µM for S100A12 [66] contradicts our chemical crosslinking data (see Section 2.3). Therefore, the SPR estimates of the *K_d_* values (Table 1) can correspond to both monomeric and dimeric forms of the S100 proteins. Meanwhile, affinities of the cytokines IL-11 and IFN-β to S100P monomer exceed those to its dimer by 1.4–2.2 orders of magnitude [36,39,40]. Hence, monomeric forms of S100A11/A12/A13 may possess even higher affinities to sTNF. 

The analogous SPR experiments for another member of a TNF-like family (SCOP [9] ID: 4000294), recombinant human TNF-β (pairwise sequence identity with human sTNF calculated using Clustal Omega 2.1 [67] is 32%), did not reveal any effects (Figure 1), thereby confirming the selective nature of the sTNF recognition by S100A11/A12/A13 proteins. Since pairwise sequence identities of S100A11/A12/A13 range from 33% to 36%, the S100-sTNF interactions are not redundant. Although S100 proteins are known to share binding partners, S100A11/A12/A13 proteins belong to the group of ‘orphan’ members of the S100 protein family, which exhibit notably higher selectivity towards binding partners compared to the other S100 proteins [68,69].

Considering that sTNF coupled to the SPR chip surface was easily regenerated after association with the S100 upon calcium depletion via passage of 20 mM EDTA solution pH 8.0 (data no shown), the conformational changes triggered in the S100 proteins by Ca^2+^ binding are important for their interaction with sTNF.

Examination of the IntAct [70] and BioGRID [71] databases shows an absence of known non-receptor extracellular proteins interacting with S100A12. Meanwhile, S100A13 was previously shown to bind cytokines IL-1α and FGF1, thereby favoring their non-canonical secretion [47,48]. Furthermore, S100A13 interacts with C-C motif chemokine 5, as evidenced by affinity purification-mass spectrometry [46]. Similarly, S100A11 was shown to bind MIF by multiplex co-fractionation/mass spectrometry [45]. Therefore, the identified S100-sTNF interactions expand the range of the cytokines potentially regulated by S100A11/A12/A13 proteins.

### 2.2. Fluorimetric Study of sTNF-S100A11/A12 Interactions

The equilibrium association constant for sTNF multimerization remains underexplored, which is probably related to the kinetic limitations due to a slow dissociation of sTNF multimer(s) with the half-life estimates in the range from 600 s [53] to 30 h [52]. Meanwhile, it was shown that the sTNF monomer is stable at a picomolar protein level, and 2 nM sTNF rapidly multimerizes, with the half-transition close to 0.1 nM [8]. To quantitate the interaction between multimeric sTNF and S100A11/A12 proteins at 25 °C, the titrations of 280 nM or 0.5 µM sTNF solution by stock solutions of Ca^2+^-bound S100A11/A12 (note that these proteins lack Trp residues) were followed by tryptophan fluorescence of sTNF (2 Trp residues; excitation at 295/298 nm) (Figure 2 and Appendix A). The titrations are accompanied by quenching of the fluorescence emission intensity of sTNF by about 25% with a bend near the equimolar ratio of S100 to sTNF, indicating binding of one S100 molecule per sTNF molecule. The titration data were described by a single site binding model (Figure 2) with an apparent equilibrium dissociation constant of (104 ± 23) nM and (348 ± 50) nM for S100A11 and S100A12 proteins, respectively. Hence, sTNF multimerization suppresses its specificity for S100A11/A12 proteins by a factor of 4/151, respectively (see Table 1).

### 2.3. Chemical Crosslinking

The interaction between sTNF and S100A11/A12/A13 proteins at 25 °C was studied using crosslinking by EDAC/sulfo-NHS, followed by the SDS-PAGE analysis (Figure 3). The cross-linked 4 µM sTNF sample exhibits presence of monomeric, dimeric and some higher-order multimeric forms. The cross-linked 4 µM S100A11 is mainly dimeric with minor contribution of the monomer under calcium excess (Figure 3A). The equimolar mixing of sTNF and Ca^2+^-free S100A11 results in the lane resembling superposition of the bands corresponding to the pure proteins. Meantime, the equimolar mixing of sTNF and Ca^2+^-bound S100A11 induces slight bleaching of the bands corresponding to monomeric and dimeric S100A11. Besides, the further 2-fold increase in S100A11 concentration does not cause accumulation of its monomer. These effects indicate formation of the sTNF-S100A11 complex in the presence of Ca^2+^. The similar, but more pronounced, effects are observed for the S100A12 protein (Figure 3B). The cross-linked 4 µM S100A12 contains monomer and dimer regardless of calcium content. The addition of 4 µM sTNF to 2 µM S100A12 under calcium excess induces disappearance of monomeric and dimeric forms of S100A12. The increase in the S100A12 concentration to 4 µM shows notable bleaching of the bands corresponding to S100A12 and sTNF dimers, and rearrangement of the bands above 50 kDa. The addition of 2-fold molar excess of Ca^2+^-bound S100A13 over sTNF leads to the decline in the intensity of the band corresponding to the sTNF monomer (Figure 3C). Thus, the crosslinking experiments evidence sTNF interaction with Ca^2+^-loaded forms of S100A11/A12/A13 proteins.

### 2.4. Influence of Extracellular sTNF and S100A11/A12/A13 on Viability of Huh-7 Cells

To probe the ability of S100A11/A12/A13 proteins to affect physiological activity of sTNF, we have studied the changes in the viability of human hepatoma Huh-7 cells induced by the addition of S100/sTNF alone or their mixture. This cell line has been reported to be sensitive to sTNF [72,73,74]. Indeed, 6 nM sTNF statistically significantly reduces the viability of Huh-7 cells compared to the untreated control by ca 12% (Figure 4A). Meanwhile, none of the studied S100 proteins (9 nM) markedly changes the viability of Huh-7 cells. Notably, the combined action of sTNF and S100A13 statistically significantly suppresses the sTNF-induced effect. This phenomenon is less prominent in the case of S100A12 (*p* = 0.051), and is absent for S100A11. Meanwhile, the minor effects, comparable with the accuracy of the experiments, raise concerns about the reliability of these conclusions.

Many of the immortalized cell lines are poorly susceptible to sTNF-induced apoptosis, which is largely ascribed to the activation of NF-κB [75,76,77,78,79]. This activation can be prevented by the use of low doses of protein synthesis inhibitors, such as CHX [74,80]. To enhance the sensitivity of the cells to sTNF, the same experiments with Huh-7 cells were performed in the presence of 10 µg/mL CHX (Figure 4B). As expected, sTNF decreases the cell viability compared to the control by ca 46%. The addition of S100A13 or S100A12 statistically significantly protects Huh-7 cells from the cytotoxic effect of sTNF, whereas S100A11 does not reveal the marked protective effect.

Taken together, these results suggest that S100A12/A13-bound forms of sTNF exhibit suppressed cytotoxic activity against Huh-7 cells. The lack of a noticeable effect in the case of S100A11 may be related to its insufficiently high affinity to sTNF (Table 1). The inhibitory action of exogenous S100 proteins on cytokine activity was also previously shown for the interaction of IFN-β with S100A1/A4/B/P proteins [36,43,49], as well as for the FGF2-S100B interaction [50].

### 2.5. Structural Modeling of the sTNF-S100 Complexes

Quaternary structures of the complexes between monomeric sTNF and S100A11/A12/A13 monomer have been predicted using the ClusPro docking server [81] (Figure 5). The predicted S100-binding sites of sTNF are located on one side of the β-sandwich (Figure 5A), and mostly consist of residues from 5 β-strands and 1–6 N-terminal residues (Figure 5C). The conserved residues of the S100-binding site of sTNF include R6, H15, N34, Y59, Y119 and Y151. The bound S100A11/A12/A13 molecules are predicted to be oriented differently with respect to the sTNF molecule (Figure 5A). In any case, the S100 residues from helices I and IV are involved in these interactions (Figure 5B). These helices along with helix III and the loop region between helices II and III (‘hinge’) are often engaged in the target recognition by S100 proteins [82]. Notably, the ‘hinge’ and helix III of S100A11/A12/A13 lack any predicted contacts with the sTNF molecule. Instead, S100A12/A13 proteins are also predicted to recognize sTNF via the residues of the non-canonical Ca^2+^-binding loop, and the N-terminal residues in the case of S100A13.

The modeling indicates that the structural details of sTNF recognition by the S100 proteins demonstrate both common and unique features compared to the predictions for the complexes of other S100 proteins with some of the four-helical cytokines. For instance, residues of the ‘hinge’, EF-loop 2 and helices III and IV of S100A1/A4/A6/P proteins were predicted to bind IFN-β [43]. Similarly, S100A2/A6/P were shown to bind EPO mostly via the residues of helices I, III and IV, and the ‘hinge’ [41]. Furthermore, S100P was predicted to recognize numerous four-helical cytokines via helices I and IV, and the ‘hinge’ [38].

The predicted S100-binding sites of sTNF intersect with the residues shown to interact with therapeutic anti-TNF monoclonal antibodies (Figure 5C). In fact, the residues R6, N34, L36 and E146 of monomeric sTNF bind Adalimumab Fab fragment (PDB entry 3WD5). Similarly, residues N34, Q61, N92, S95, Y119 and E146 of trimeric sTNF interact with Infliximab Fab fragments (PDB entry 4G3Y) (Figure 5C). Therefore, S100A11/A12/A13 binding to sTNF could interfere with the anti-TNF therapy based on use of these antibodies.

The residue E146 of sTNF, predicted to bind S100A13 (Figure 5C), has previously been shown to be involved in trimeric sTNF recognition of the TNF-R1 and TNF-R2 receptors (see PDB entries 7KPB and 3ALQ, respectively). In addition, the sTNF residues R32/A33 and Q149 able to interact with the TNF receptors (PDB entries 7KPB, 3ALQ) are located nearby the residues N34 and Y151, predicted to bind S100A11/A12/A13 proteins (Figure 5C). The partial overlap of the sTNF residues predicted to constitute the S100-binding sites with the residues recognizing the TNF-R1/R2 receptors is consistent with the inhibitory effects of S100A12/A13 on the cytotoxic activity of sTNF against Huh-7 cells (Figure 4B).

### 2.6. Human Diseases Associated to Dysregulation of TNF and S100A11/A12/A13

To reveal the diseases associated with the simultaneous involvement of TNF and either the S100A11, S100A12 or S100A13 protein, we analyzed the human disease databases v7.0 [86] and Open Targets Platform v.22.09 (‘OTP’) [87] as described in ref. [36].

DisGeNET contains 92 entries related to both TNF and S100A11, which mostly correspond to the various neoplasms (Appendix A). The OTP database contains 146 entries associated with both TNF and S100A11 (Appendix A), but the association scores exceed 0.1 only in the case of asthma.

DisGeNET contains 109 diseases related to the dysregulation of both TNF and S100A12, including numerous neoplasms, arthritis, diabetes and many other maladies (Appendix A). The OTP database includes 252 entries associated with both TNF and S100A12 (Appendix A), but only inflammatory bowel disease has the association scores above 0.1.

Although the analysis of the DisGeNET database does not reveal the entries related to both TNF and S100A13, the OTP database includes 42 such entries (Appendix A), but none of them has the association scores exceeding 0.1.

Analysis of the OTP database shows that 13 entries are simultaneously related to the TNF and S100A11/A12/A13 (Appendix A), including various neoplasms, diabetes, coronary artery disease and gonorrhea. Meanwhile, none of the entries has the association scores above 0.1. Despite the low association scores in some cases, the examination of the DisGeNET and OTP databases indicates that the TNF-S100 interactions potentially mediate the progression of multiple disorders.

## 3. Materials and Methods

### 3.1. Materials

Recombinant human sTNF (corresponding to the residues 77–233 of the UniProt entry P01375 with an additional N-terminal methionine) produced in *E. coli* was bought from Prospec-Tany Technogene Ltd. (Ness-Ziona, Israel) (catalog # CYT-223). Alternatively, recombinant human sTNF with an N-terminal hexahistidine tag from ThermoFisher Sci. (Waltham, MA, USA) (catalog # RP-75738) was used for MTT assay. Recombinant human TNF-β (UniProt entry P01374) produced in *E. coli* was from BioVision Inc. (Zurich, Switzerland) (catalog number 4345). Recombinant human S100A1/A2/A3/A4/A5/A6/A7/A8/A9/A10/A11/A12/A13/A14/A15/A16/B/P proteins were expressed in *E. coli* and purified as previously described [40,41,43,51]. Protein concentrations were determined spectrophotometrically according to ref. [88].

HEPES, Tris, glycine, sodium chloride, sodium hydroxide, DTT and SDS were from PanReac AppliChem. CaCl_2_, EDTA, TWEEN 20, CHX and DMSO were from Sigma-Aldrich Co. (Burlington, MA, USA) ProteOn™ GLH sensor chip, amine coupling kit, EDAC and sulfo-NHS were from Bio-Rad Laboratories, Inc. (Hercules, CA, USA) NAP-5 column was from Cytiva (Marlborough, MA, USA).

Human hepatoma Huh-7 cell line was kindly provided by Eliseeva I.A. (Institute of Protein Research, Russia). DMEM, penicillin, streptomycin and L-glutamine were bought from PanEco Ltd. (Moscow, Russia). Fetal bovine serum was from Biosera. MTT was purchased from Dia-M (Moscow, Russia).

### 3.2. Surface Plasmon Resonance Analysis

SPR studies of sTNF or TNF-β interaction with S100 proteins at 25 °C were performed using the ProteOn™ XPR36 protein interaction array system (Bio-Rad Laboratories, Inc., Hercules, CA, USA). Ligand (50 μg/mL sTNF or TNF-β) in 10 mM sodium acetate pH 4.5 buffer was immobilized on the surface of a ProteOn™ GLH sensor chip (up to 13,000 RUs) by amine coupling, followed by blocking of the remaining activated amine groups on the chip surface by 1 M ethanolamine solution. The chip surface was flushed with the running buffer (10 mM HEPES, 150 mM NaCl, 1 mM CaCl_2_, 0.05% TWEEN 20, pH 7.4) for two days and several times with 10 mM glycine pH 3.3 for 150 s to ensure dissociation of sTNF multimer(s) [52,53]. The analyte (S100 protein, 10 nM–2 µM) in the running buffer was passed over the chip at a rate of 30 μL/min for 300 s, followed by flushing the chip with the running buffer for 2400 s. The sensor chip surface was regenerated by passage of 20 mM EDTA pH 8.0 solution for 300 s.

The double-referenced SPR sensograms were described within a heterogeneous ligand Model (1), which suggests the existence of two populations of the ligand (L_1_ and L_2_) that bind a single analyte molecule (A):
(1)$$\begin{array}{ccccc} & k_{a1} & & k_{a2} & \\ L_{1}+A & ⇄ & L_{1}A\;\;\;\;\;\;\;\;\;\;L_{2}+A & ⇄ & L_{2}A\\ & k_{d1} & & k_{d2} & \\ & K_{d1} & & K_{d2} & \end{array}$$
where *K_d_* and *k_d_* correspond to equilibrium and kinetic dissociation constants, respectively. The ligand may be exposed to the analyte in various preferential conformations, corresponding to different analyte-binding sites or different conformations of the same site. Thus, L_1_ and L_2_ correspond to “open” conformations of the ligand, which differ in their specificity to the analyte. *K_d_* and *k_d_* values were calculated for each analyte concentration using ProteOn Manager™ v.3.1 software (Bio-Rad Laboratories, Inc.); the averaged values and standard deviations (n = 3–4) are indicated.

### 3.3. Fluorescence Studies

Fluorescence measurements were performed using a Cary Eclipse spectrofluorometer (Varian, Inc., Palo Alto, CA, USA). sTNF solution in 10 mM HEPES-NaOH, 150 mM NaCl, 1 mM CaCl_2_, pH 7.4 buffer in a 10 × 4 mm quartz cell was titrated at 25 °C by stock solution of S100A11 or S100A12 in the same buffer. sTNF concentration was 280 nM or 0.5 µM for the titrations by S100A11 and S100A12, respectively. Since, contrary to sTNF, S100A11/A12 proteins lack Trp residues, an excitation wavelength of 295/298 nm (excitation and emission monochromator bandwidths were 2.5 nm and 10 nm, respectively) ensured selective detection of fluorescence emission spectra of Trp residues of sTNF. The fluorescence spectra were corrected for spectral sensitivity of the fluorimeter and fitted by a log-normal function [89], using LogNormal software (IBI RAS, Pushchino, Russia). The dependence of fluorescence intensity of sTNF at 325 nm on relative concentration of a S100 protein was fitted by a single site binding model using FluoTitr software (IBI RAS, Pushchino, Russia), which implements Marquardt algorithm [90] and takes into consideration the dilution effect [91].

### 3.4. Chemical Crosslinking

Crosslinking of sTNF with S100A11/A12/A13 proteins was performed mainly as described in ref. [49]. The S100 protein (1 mg/mL solution in 20 mM Tris-HCl, 20 mM DTT, 20 mM EDTA, pH 8.0 buffer) was loaded onto a NAP-5 desalting column and washed with buffer A (10 mM HEPES, 150 mM NaCl, pH 7.4) to remove calcium. The protein samples (4 µM sTNF with the S100 added up to sTNF to S100 molar ratio of 1:0, 0:1, 1:0.5, 1:1, 1:2) were treated with 20 mM EDAC and 5 mM sulfo-NHS at 25 °C (buffer A with/without 1 mM CaCl_2_). The reaction was quenched after 1.5 h by addition of SDS-PAGE sample loading buffer, followed by SDS-PAGE (15%) with silver staining and the gel scanning using a Molecular Imager PharosFX Plus System (Bio-Rad Laboratories, Inc., Hercules, CA, USA).

### 3.5. Cell Viability Studies

Huh-7 cells were grown in DMEM supplemented with 10% fetal bovine serum, 100 U/mL penicillin, 100 μg/mL streptomycin and 2 mM l-glutamine. One hundred µL of cell suspension (5 × 10^4^ cells) were seeded in quadruplicates into 96-well plates, and 10 ng of sTNF/S100A11/A12/A13 were added to the cells either alone or in combination of sTNF with one of the S100 proteins (final protein concentrations of 6 nM and 9 nM for sTNF and S100, respectively). Where indicated, CHX was added to the cells to a final concentration of 1 μg per well. The cells were grown at 37 °C in a humidified 5% CO_2_ atmosphere for 16 h. 10 μL of 5 mg/mL MTT stock solution were added to the cells, and the plates were incubated under the same conditions for 3 h. The medium was carefully removed and 100 µL of DMSO per well were added to dissolve the formazan crystals. The absorbance at 590 nm was measured using a FilterMax F5 microplate reader (ThermoFisher Sci.). The experiments were repeated at least four times. The quantitative results are presented as mean +/− standard deviation. The differences analyzed using Student’s *t*-test were considered significant at *p*-values below 0.05.

### 3.6. Structural Modeling of sTNF-S100 Complexes

The models of quaternary structures of sTNF-S100A11/A12/A13 complexes were built using ClusPro docking server (https://cluspro.bu.edu/login.php, accessed on 1 November 2022) [81]. The tertiary structure of human sTNF monomer was extracted from the crystal structure of its complex with an Adalimumab Fab fragment (chain A of PDB [83] entry 3WD5). The structures of monomeric forms of S100A11/A12/A13 proteins were predicted using AlphaFold (https://alphafold.ebi.ac.uk/, accessed on 1 November 2022) [84]. Distributions of the contact residues in the docking models over the protein sequences were calculated using Python 3.3 programming language as implemented in PyCharm v.3.0.2 development environment, Matplotlib Python plotting library and NumPy numerical mathematics extension. The residues that are contact residues in the five or more docking models represent the most probable binding site. The models that most closely match the probable binding sites were drawn using PyMOL v.1.6.9.0 software (https://pymol.org/2/, accessed on 1 November 2022).

### 3.7. Search of the Diseases Associated with the TNF and S100 Proteins

The data on diseases associated with human TNF (UniProt ID P01375) and either S100A11 (UniProt ID P31949), S100A12 (UniProt ID P80511) or S100A13 (UniProt ID Q99584) were collected from the human disease databases DisGeNET v7.0 (https://www.disgenet.org/, accessed on 1 November 2022) [86] and Open Targets Platform v.22.09 (https://platform.opentargets.org/, accessed on 1 November 2022) [87] as described in ref. [36]. The DisGeNET entries were manually curated; false positive records were removed.

## 4. Conclusions

The interactions between monomeric/multimeric sTNF and Ca^2+^-bound forms of S100A11/A12/A13 proteins in vitro are demonstrated here using SPR spectroscopy, intrinsic fluorescence, and chemical crosslinking techniques. These interactions are not redundant and are selective, as confirmed by their disappearance for TNF-β and 15 other S100 proteins. The lowest values of the respective dissociation constants reach the nanomolar levels, which are comparable to the concentrations of extracellular forms of S100A11/A12/A13 under (patho)physiological conditions. This fact indicates a possible regulatory role for the sTNF-S100 interactions, which is further supported by the cell viability analysis using Huh-7 human hepatoma cells as a model system.

We have shown that exogenous S100A12/A13 rescue Huh-7 cells from the cytotoxic effect of sTNF. The relevance of this phenomenon for treatment of liver cancer remains unclear due to the controversial role of TNF in cancer, exerting both pro- and anticarcinogenic effects [92]. TNF promotes hepatocarcinogenesis via activation of hepatic progenitor cells [93]. Further, the induction of apoptosis may cause formation of abnormal cells in the process of regeneration and dysplasia, leading to oncogenesis [94]. Meanwhile, TNF exhibits antitumor activity by acting as an anti-angiogenic factor, increasing vascular permeability in tumors, and enhancing the response of HCC to anti-cancer agents. In addition, high expression of TNF and NF-κB in tumor microenvironment promotes survival in HCC [95]. Similarly, S100A11/A12/A13 proteins are known to be involved into hepatocarcinogenesis [96]. S100A12 expression in HCC is restricted to stromal cells, and overexpression of S100A12 on intratumoral stromal cells predicts poor prognosis after surgical resection [97]. However, S100A12 expression levels in HCC tissues are lowered, and elevated S100A12 expression correlates with favorable overall survival [96]. At the same time, S100A13 expression levels are elevated, which is associated with a poor overall survival. Besides, the expression levels of S100A11 and S100A13 in HCC are associated with infiltrates of B/T cells, macrophages, neutrophils, and dendritic cells (only macrophages and neutrophils in the case of S100A12) [96], known to express TNF [5]. This fact indicates the possible crosstalk between TNF and the S100 proteins secreted either by stromal (S100A12) or tumor (S100A11/A13) cells. The revealed interactions between sTNF and S100A12/A13 are shown to affect viability of the HCC cells, thereby pointing out that modulation or consideration of these interactions could serve as a treatment option for liver malignancies. The same considerations hold true for the many other disorders associated with simultaneous dysregulation of TNF and S100A11/A12/A13 identified by the bioinformatic analysis.

The structural modelling of the sTNF-S100 complexes predicts an overlap of the S100-binding sites of sTNF with the epitopes recognized by the therapeutic anti-TNF monoclonal antibodies. Therefore, the S100 binding to sTNF could affect the therapeutic efficacy of anti-TNF antibodies, which may partly explain the fact that not all patients respond to the TNF inhibitors [1]. This suggestion needs further verification as it opens up novel opportunities for personalization of the anti-TNF therapies. For instance, serum S100A12 levels have been associated with response to anti-TNF therapy in patients with juvenile idiopathic arthritis and Crohn’s disease [57,98].

While S100A12 lacks reported non-receptor extracellular binding partners, S100A13 has previously been shown to bind IL-1α, FGF1 and C-C motif chemokine 5 [46,47,48], and S100A11 has been reported to interact with MIF [45]. Hence, the novel sTNF-S100A11/A12/A13 interactions extend our knowledge of the regulatory activity of these S100 proteins with regard to cytokines. These interactions are particularly valuable given that S100A11/A12/A13 are much more selective for binding partners than other S100 proteins [68,69]. Some of other members of the S100 protein family proteins interact with the numerous four-helical cytokines: IL-3/5/9/10/13/15/20/21/22/24/26/27/31/35, G-CSF, GM-CSG, GH, GH-V, PRL, LEP, THPO, IFN-ω1 [38], IL-11, OSM, CNTF, CT-1, CLCF1 [40,42,51], IFN-β [36,43,49] and EPO [41]. The specificity of S100 proteins to other representatives of the TNF-like family (SCOP [9] ID: 4000294) needs further exploration. Meanwhile, the available data allow us to put forward a hypothesis about specialization of certain subsets of S100 proteins in regulation of specific cytokines belonging to different structural/functional families, thereby establishing the next complexity level in regulation of their activities.

## Figures and Tables

**Figure 1 ijms-23-15956-f001:**
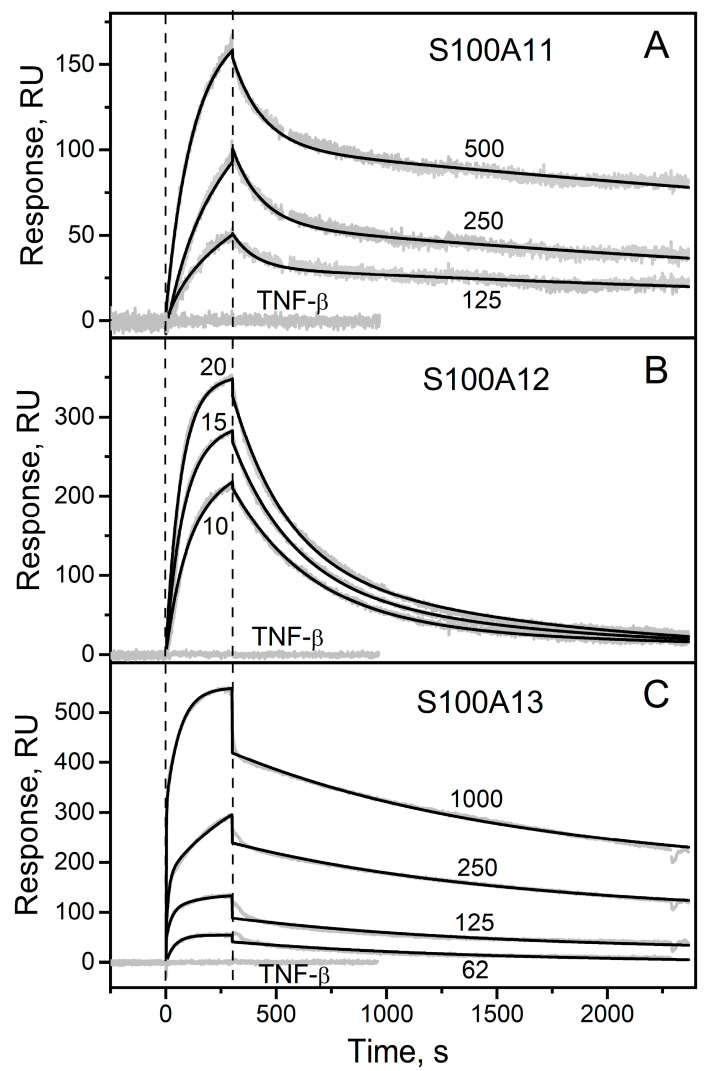
Kinetics of the interaction of sTNF with Ca^2+^-loaded proteins S100A11 (panel (**A**)), S100A12 (**B**) or S100A13 (**C**) at 25 °C, monitored by SPR spectroscopy (10 mM HEPES, 150 mM NaCl, 1 mM CaCl_2_, 0.05% TWEEN 20, pH 7.4). sTNF is immobilized on the sensor chip surface by amine coupling. The S100 protein concentrations in nM are indicated for the sensograms. The sensograms for TNF-β used as a ligand and S100 protein concentration of 2 µM are shown. The grey curves are experimental, while the black curves are calculated according to the heterogeneous ligand model (see Section 3.2) (see Table 1). The vertical dotted lines mark the association phase, followed by the dissociation phase.

**Figure 2 ijms-23-15956-f002:**
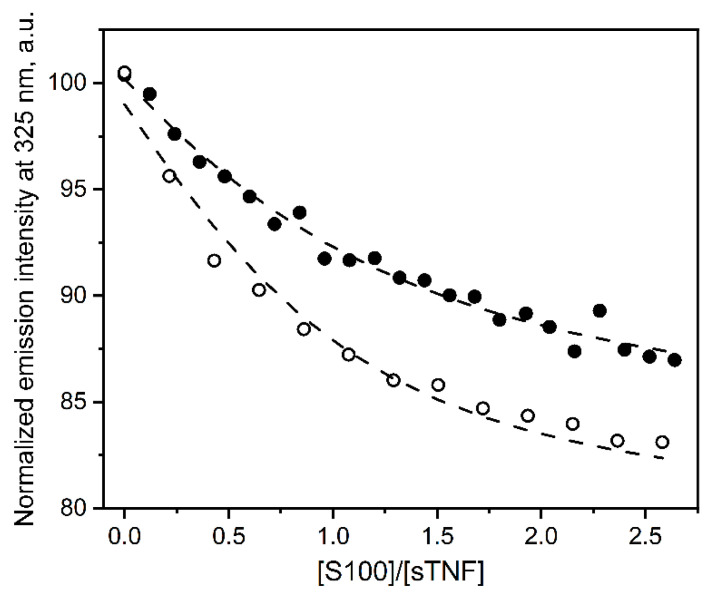
The interaction of sTNF with Ca^2+^-bound S100A11 (open circles) or S100A12 (solid circles) at 25 °C, followed by fluorescence emission of Trp residues of sTNF (10 mM HEPES-NaOH, 150 mM NaCl, 1 mM CaCl_2_, pH 7.4). sTNF concentration of 280 nM or 0.5 µM for the titrations by S100A11 and S100A12, respectively. The excitation wavelength was 295/298 nm. The circles denote the experimental data, while the dashed curves are fits according to a single site binding model.

**Figure 3 ijms-23-15956-f003:**
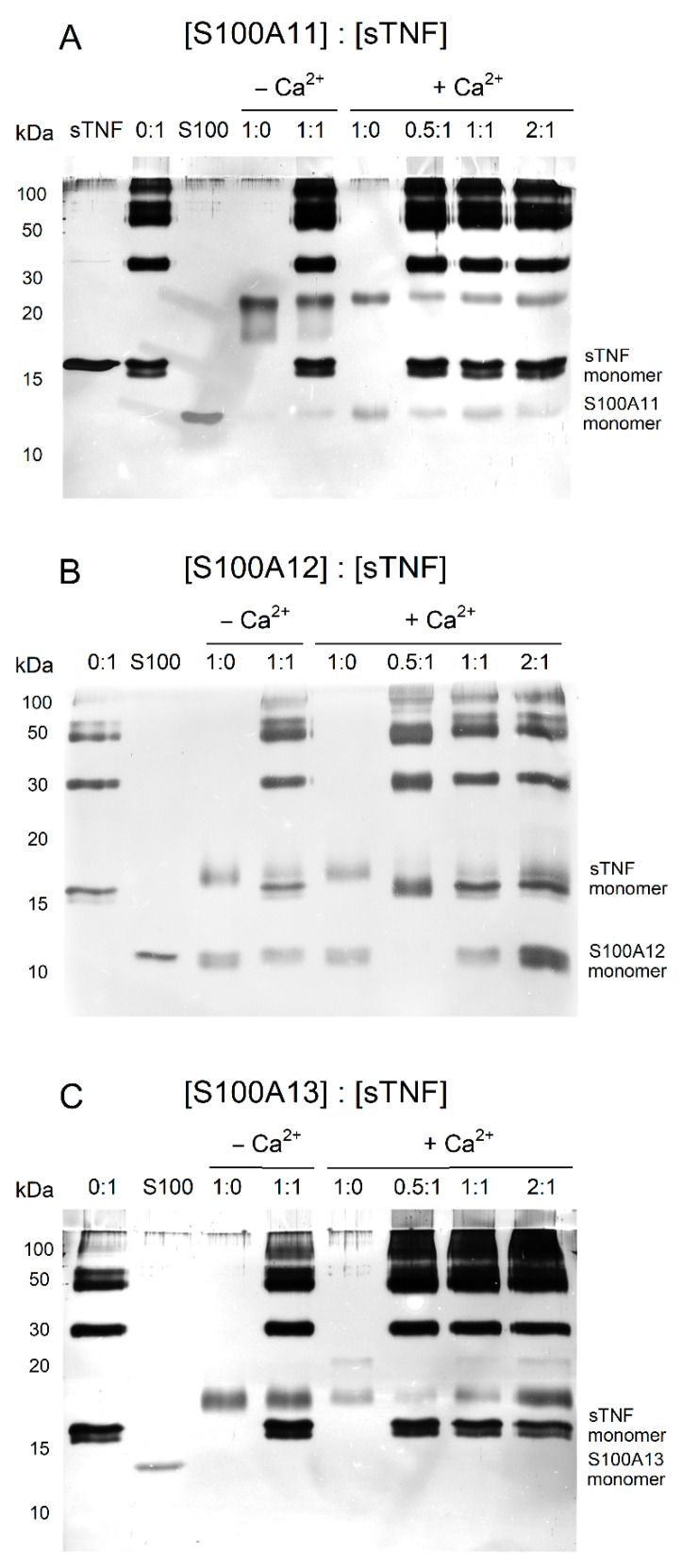
The results of SDS-PAGE for 4 µM sTNF with S100A11 (panel (**A**)), S100A12 (**B**) or S100A13 (**C**) (S100 to sTNF molar ratios are indicated), after crosslinking with EDAC/sulfo-NHS at 25 °C for 1.5 h (10 mM HEPES, 150 mM NaCl, pH 7.4 buffer with/without 1 mM CaCl_2_). The control lanes for the proteins without crosslinking are marked as ‘sTNF’ and ‘S100’.

**Figure 4 ijms-23-15956-f004:**
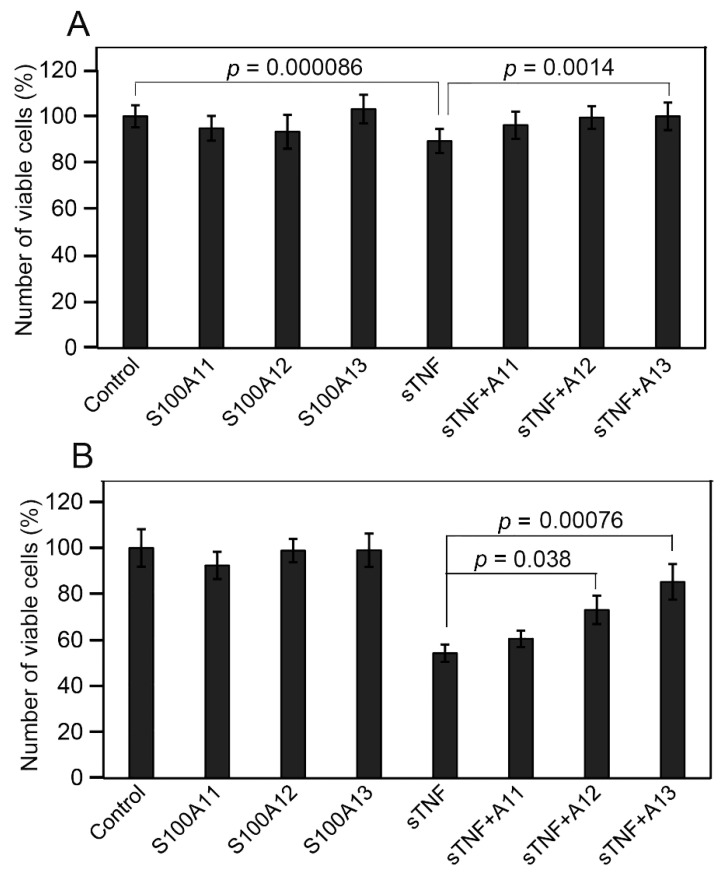
The effect of extracellular sTNF/S100A11/A12/A13 (6 nM and 9 nM for sTNF and S100, respectively) and their combinations on the viability of Huh-7 cells, measured by MTT assay. The untreated cells served as a control. (**A**) The cells were cultured with the S100 proteins either in the absence or presence of sTNF, as indicated. (**B**) The same experiments performed in the presence of 10 µg/mL CHX to block the sTNF-induced survival pathways. The data represent the mean values from at least four independent experiments performed in quadruplicates. The errors correspond to standard deviations. The *p*-values were calculated according to Student’s *t*-test.

**Figure 5 ijms-23-15956-f005:**
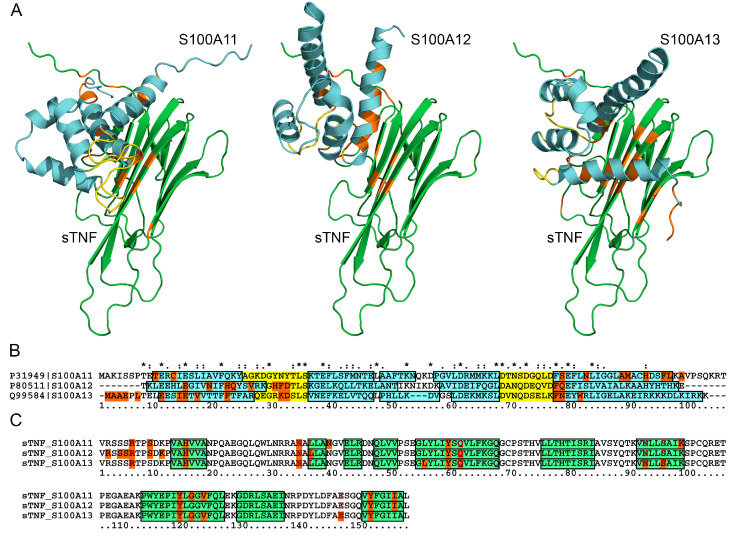
(**A**) The models of quaternary structures of the complexes between monomeric sTNF (taken from PDB [83] entry 3WD5; shown in green) and S100A11/A12/A13 monomer (predicted using AlphaFold [84]; shown in cyan), built using the ClusPro docking server [81]. The Ca^2+^-binding loops and the contact residues are shown in yellow and orange, respectively. The same color scheme is used in the panels (**B**,**C**). (**B**) Mapping of the predicted contact residues of the S100 proteins onto their aligned amino acid sequence (Clustal Omega algorithm [85]). The helical regions are marked by the cyan boxes. The identical and similar residues are marked with “*” and “.”/”:”, respectively. (**C**) The predicted contact residues of sTNF for its complexes with S100A11/A12/A13. The beta strands are marked by the green boxes according to PDB entry 3WD5. The contact residues of sTNF involved in interaction with Adalimumab Fab fragment (PDB entry 3WD5) or Infliximab Fab fragments (PDB entry 4G3Y) are highlighted in bold red.

**Table 1 ijms-23-15956-t001:** Parameters of the heterogeneous ligand model (see Section 3.2), describing the SPR data on the kinetics of interaction between sTNF and S100 proteins at 25 °C, shown in Figure 1.

Analyte	*k_d_*_1_, s^−1^	*K_d_*_1_, nM	*k_d_*_2_, s^−1^	*K_d_*_2_, nM
S100A11	(1.8 ± 0.5) × 10^−4^	28 ± 16	(8.4 ± 0.3) × 10^−3^	695 ± 73
S100A12	(1.8 ± 1.4) × 10^−3^	2.3 ± 1.7	(2.3 ± 1.7) × 10^−3^	8.4 ± 0.5
S100A13	(9.0 ± 1.6) × 10^−4^	2.4 ± 0.7	(7.3 ± 2.5) × 10^−3^	4.3 ± 2.9

## Data Availability

The data supporting the reported results can be found at the Laboratory of New Methods in Biology of the Institute for Biological Instrumentation, Pushchino Scientific Center for Biological Research of the Russian Academy of Sciences, 142290 Pushchino, Russia.

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
