# Peer review of "Specific S100 Proteins Bind Tumor Necrosis Factor and Inhibit Its Activity"

_ijms, 2022, doi:10.3390/ijms232415956_

Round 1

Reviewer 1 Report

The manuscript by Kazakov and colleagues reports on the interaction between S100 proteins and TNF. Overall the manuscript is highly interesting and represent a good contribution to the field. I would only suggest the authors to better explain the reason why the authors opted to check the interaction between S100 proteins and TNF since it is not very clear from the introduction.

Author Response

The manuscript by Kazakov and colleagues reports on the interaction between S100 proteins and TNF. Overall the manuscript is highly interesting and represent a good contribution to the field. I would only suggest the authors to better explain the reason why the authors opted to check the interaction between S100 proteins and TNF since it is not very clear from the introduction.

RESPONSE: This choice was largely based on prior knowledge of the extensive interactions between several S100 proteins and specific cytokines, including those belonging to the all-beta class (the case of TNF). In addition, S100 proteins and TNF significantly overlap in the diseases associated with their disturbance, which indicates the presence of direct or indirect relationships between them. We have included these considerations in the last paragraph of the introduction: “The extensive interactions between several S100 proteins and specific cytokines, including those belonging to the all-beta class (the case of TNF), and a significant overlap in the diseases associated with disturbance of S100 proteins or TNF, indicate the possibility of direct or indirect relationships between them”.

Reviewer 2 Report

The article describes interactions between sTNF and S100 proteins. The authors obtained interesting and scientifically sound results (for the most part). The article is well written

But there are points that need to be clarified.

1) The title "Binding of specific S100 proteins inhibits tumor necrosis factor activity" is incorrect.

The authors proved binding in vitro and showed that S100A12/A13 proteins suppress the cytotoxic activity of TNF against Huh-7 cells, but various mechanisms for this suppression can be proposed.

More correct title «Specific S100 proteins bind tumor necrosis factor and inhibit its activity”.

2) "heterogeneous ligand model (1)" (line 127 and heading of table 1). Please reference to the "Materials and Methods" section and remove italic.

3) Please explain what the two ligand populations (L1 and L2) are after the first reference to Table 1 or below the Table (as keys).

4) Line 160-162. Why the contradiction with the chemical crosslinking data discussed there?

5) The fluorimetric study is not well presented.

(5.1) The initial Trp-fluorescence quenching spectra should be presented in Fig. 2 together with their graphical analysis.

(5.2) Please check the Y-axis legend.

The fluorescence intensity cannot be the same for 280 and 0.5 nM protein.

Moreover, I am very surprised that Trp fluorescence of the 0.5 nM solution of the protein with 2 Trp is suitable for measurement.

6) Fig. 3.

To be honest, gels do not provide unequivocal evidence of complex formation (both proteins are multimerized and anything can be predicted). I would like the authors to show with arrows where, in their opinion, these complexes appear. In addition, a mass spectrometric analysis of the cut bands could give an unambiguous answer. Can't it be done?

7) Supplementary tables with PubMed identifiers of references confirming protein-disease associations for TNF and S100.

Do any of the many references give any cross information about TNF and S100? Have the authors read any of these articles? If not, why is this being presented?

Author Response

Reviewer #2

The article describes interactions between sTNF and S100 proteins. The authors obtained interesting and scientifically sound results (for the most part). The article is well written

But there are points that need to be clarified.

1) The title "Binding of specific S100 proteins inhibits tumor necrosis factor activity" is incorrect.

The authors proved binding in vitro and showed that S100A12/A13 proteins suppress the cytotoxic activity of TNF against Huh-7 cells, but various mechanisms for this suppression can be proposed.

More correct title «Specific S100 proteins bind tumor necrosis factor and inhibit its activity”.

RESPONSE: We agree that the title proposed by the reviewer is stricter, so the title has been corrected.

2) "heterogeneous ligand model (1)" (line 127 and heading of table 1). Please reference to the "Materials and Methods" section and remove italic.

RESPONSE: Done.

3) Please explain what the two ligand populations (L1 and L2) are after the first reference to Table 1 or below the Table (as keys).

RESPONSE: We have added the following explanation of the nature of L1 and L2 to the section 3.2:

“The ligand may be exposed to the analyte in various preferential conformations, corresponding to different analyte-binding sites or different conformations of the same site. Thus, L1 and L2 correspond to “open” conformations of the ligand, which differ in their specificity to the analyte”.

4) Line 160-162. Why the contradiction with the chemical crosslinking data discussed there?

RESPONSE: The chemical crosslinking data are discussed here, since they are critical to understanding the SPR data presented in this section. We have changed the link to section 2.3.

5) The fluorimetric study is not well presented.

(5.1) The initial Trp-fluorescence quenching spectra should be presented in Fig. 2 together with their graphical analysis.

RESPONSE: Since these spectra are not representative, we have shown them in the supplementary Figure S1.

(5.2) Please check the Y-axis legend.

The fluorescence intensity cannot be the same for 280 and 0.5 nM protein.

Moreover, I am very surprised that Trp fluorescence of the 0.5 nM solution of the protein with 2 Trp is suitable for measurement.

RESPONSE: In fact, the protein concentration in this experiment was 0.5 mcM, but not 0.5 nM. We have corrected this typo throughout the text of the manuscript.

The fluorescence emission intensity values shown in Figure 2 were normalized to ensure the initial intensity values of 100 arbitrary units. We have corrected the title of the Y axis.

6) Fig. 3.

To be honest, gels do not provide unequivocal evidence of complex formation (both proteins are multimerized and anything can be predicted). I would like the authors to show with arrows where, in their opinion, these complexes appear. In addition, a mass spectrometric analysis of the cut bands could give an unambiguous answer. Can't it be done?

RESPONSE: Indeed, our experience in exploration of the interactions between S100 proteins and specific cytokines (extensively referenced in the manuscript text) clearly shows that chemical crosslinking data are typically much less clear compared to the data from other techniques. In fact, formation of the S100-cytokine complexes is mostly followed by disappearance of the electrophoretic bands, corresponding to the S100 protein, cytokine, or both. The same situation is observed in the present study.

Characterization of the cross-linked products by MS is complicated due to formation of multiple inter- and intramolecular bonds upon EDAC/sulfo-NHS cross-linking, leading to high heterogeneity of the protein sample.

7) Supplementary tables with PubMed identifiers of references confirming protein-disease associations for TNF and S100.

Do any of the many references give any cross information about TNF and S100? Have the authors read any of these articles? If not, why is this being presented?

RESPONSE: The PubMed identifiers are provided in the supplementary tables to ensure the possibility of manual verification of the bioinformatically identified associations between TNF/S100 dysregulation in specific diseases.

Reviewer 3 Report

In this study, the authors have investigated the interaction of eighteen representatives of the multifunctional S100 protein family proteins with Tumor necrosis factor (TNF) using surface plasmon resonance spectroscopy. Further, the authors have performed some in vitro experiments and bioinformatics analysis and implicated that sTNF(soluble)-S100 interactions may interfere with the sTNF recognition by the therapeutic anti-TNFs. While the study is interesting, it has a few concerns regarding in vitro experiments and bioinformatics analysis.

1.     Fig 4A – The cell viability percentage among different samples is slightly increased or decreased. Besides, S100A11 and S100A12 treated cells showed less cell viability than control cells. In fact, S100A11 and S100A12 treated cells were similar to sTNF-treated cells in terms of cell viability. Of note, S100A13 alone increased cell viability, which is comparable to sTNF and S100A13 treated cells. Hence, the observed effect could be independent of sTNF suppression.

2.     Fig 4B – Though the authors have used cycloheximide to inhibit the activation of NF-κB by sTNF, cell viability assay alone is not enough to conclude that S100 proteins suppress the sTNF-mediated cell cytotoxicity. Could authors demonstrate through additional experiments (for example, using western blot or other techniques) by quantifying apoptotic proteins that S100A13 rescue Huh-7 cells from sTNF-mediated cell death?

3.     Here, the authors postulate that S100 proteins interact with sTNF, thereby it inhibits its activity. In this context, what about the sTNF interaction with TNF receptors? Do the predicted S100-binding sites of sTNF overlap with residues/motifs that are important for sTNF interaction with TNF receptors?

Minor comments

Line 243- Is it 10g/ml of Cycloheximide?

Line 408- Typo

Author Response

In this study, the authors have investigated the interaction of eighteen representatives of the multifunctional S100 protein family proteins with Tumor necrosis factor (TNF) using surface plasmon resonance spectroscopy. Further, the authors have performed some in vitro experiments and bioinformatics analysis and implicated that sTNF(soluble)-S100 interactions may interfere with the sTNF recognition by the therapeutic anti-TNFs. While the study is interesting, it has a few concerns regarding in vitro experiments and bioinformatics analysis.

  1. Fig 4A – The cell viability percentage among different samples is slightly increased or decreased. Besides, S100A11 and S100A12 treated cells showed less cell viability than control cells. In fact, S100A11 and S100A12 treated cells were similar to sTNF-treated cells in terms of cell viability. Of note, S100A13 alone increased cell viability, which is comparable to sTNF and S100A13 treated cells. Hence, the observed effect could be independent of sTNF suppression.

RESPONSE: Indeed, the effects shown in Figure 4A are small and comparable to accuracy of the MTT assay. This was the reason for the use of cycloheximide to block pro-survival pathways and thereby enhance the cellular effects (see Figure 4B). In any case, we described only statistically significant effects. Although there is a slight trend towards decreased cell viability in response to S100A11 or S100A12, these effects did not reach statistical significance. Similarly, the S100A13-induced increase in the cell viability (+3.1%) did not reach statistical significance with p value of 0.38. The observed inhibitory effect of S100A13 on sTNF activity was qualitatively confirmed by the experiment with addition of cycloheximide (see Figure 4B).

In order to emphasize the limited reliability of the conclusions from the experiments carried out in the absence of cycloheximide, the following sentence was added at the very end of the first paragraph of section 2.4:

“Meanwhile, the minor effects, comparable with accuracy of the experiments, raise concerns about reliability of these conclusions”.

  1. Fig 4B – Though the authors have used cycloheximide to inhibit the activation of NF-κB by sTNF, cell viability assay alone is not enough to conclude that S100 proteins suppress the sTNF-mediated cell cytotoxicity. Could authors demonstrate through additional experiments (for example, using western blot or other techniques) by quantifying apoptotic proteins that S100A13 rescue Huh-7 cells from sTNF-mediated cell death?

RESPONSE: We respectfully disagree with the notion that the MTT assays presented in the manuscript are insufficient to assess influence of the exogenous S100 proteins on the cytotoxic activity of sTNF. The MTT assay evaluates the content of metabolically active cells and thereby discriminates viable cells from a population of dead cells. Indeed, the additional information on the cell death mechanisms behind the observed effects could provide valuable information, but these details are beyond the scope of the present study. In fact, we are planning to carry out for the Huh-7 cells treated with sTNF, S100 proteins and their mixtures Annexin V/PI cell death assay (evaluation of the levels of apoptosis and necrosis), analysis of sTNF-induced activation of JNK and caspase-3 pathways, and quantitation of expression of pro-inflammatory cytokines. However, these data do not affect the key message of the manuscript that specific S100 proteins interact with sTNF and exert functionally relevant effects on its signaling. We believe that the detailed study of the cellular mechanisms mediating these effects deserves a separate article.

  1. Here, the authors postulate that S100 proteins interact with sTNF, thereby it inhibits its activity. In this context, what about the sTNF interaction with TNF receptors? Do the predicted S100-binding sites of sTNF overlap with residues/motifs that are important for sTNF interaction with TNF receptors?

RESPONSE: To address this question, we have added the following considerations to the section 2.5:

“The residue E146 of sTNF, predicted to bind S100A13 (Figure 5C), has previously been shown to be involved in trimeric sTNF recognition of the TNF-R1 and TNF-R2 receptors (see PDB entries 7KPB and 3ALQ, respectively). In addition, the sTNF residues R32/A33 and Q149 able to interact with the TNF receptors (PDB entries 7KPB, 3ALQ) are located nearby the residues N34 and Y151, predicted to bind S100A11/A12/A13 proteins (Figure 5C). The partial overlap of the sTNF residues predicted to constitute the S100-binding sites with the residues recognizing the TNF-R1/R2 receptors is consistent with the inhibitory effects of S100A12/A13 on the cytotoxic activity of sTNF against Huh-7 cells (Figure 4B)”.

Minor comments:

Line 243- Is it 10g/ml of Cycloheximide?

RESPONSE: This experiment was carried out in the absence of cycloheximide. In the experiments with the addition of cycloheximide, its concentration was 10 mcg/ml. We have corrected this typo all over the manuscript text.

Line 408- Typo

RESPONSE: Corrected.

Round 2

Reviewer 3 Report

The authors have addressed all my queries. 

I agree with the author's explanation.